# Acoustically Stimulated Charge Transport in Graphene Film

**DOI:** 10.3390/nano12244370

**Published:** 2022-12-07

**Authors:** Dmitry Roshchupkin, Oleg Kononenko, Rashid Fakhrtdinov, Evgenii Emelin, Alexander Sergeev

**Affiliations:** 1Institute of Microelectronics Technology and High Purity Materials Russian Academy of Sciences, 142432 Chernogolovka, Russia; 2Federal State Institution, Scientific Research Institute for System Analysis of the Russian Academy of Sciences, 117218 Moscow, Russia

**Keywords:** graphene, surface acoustic waves, scanning electron microscopy, charge transport

## Abstract

The process of acoustically stimulated charge transport in the graphene film on the surface of the YZ−cut of a LiNbO_3_ crystal was investigated. It was found that the dependence of the current in the graphene film on the frequency of the surface acoustic wave (SAW) excitation repeats the amplitude-frequency response of the SAW delay time line. It is shown that increasing the SAW amplitude leads to an increase in the current in the graphene film, and the current in the graphene film depends linearly on the amplitude of the high-frequency input signal supplied to the interdigital transducer (IDT, in dB). It is demonstrated that at a positive bias potential on the graphene film, the SAW propagation allows to change the direction of the current in the graphene film by changing the amplitude of the SAW. It is also shown that in the frequency range of the amplitude-frequency response of the SAW delay time line, the current in the graphene film can vary from positive to negative values depending on the frequency. The capability to control the SAW excitation frequency or the SAW amplitude makes it possible to control the value and direction of the current in the graphene film. The SAW propagation lets to collect and transport the photo-stimulated charges in the graphene film on the crystal surface.

## 1. Introduction

Modern acoustoelectronics is based on the use of surface and bulk acoustic waves for the transmission and processing of information in a real time mode [1]. Also, acoustoelectronic devices are actively used as physical quantity sensors, the principle of operation of which is based on the change of resonance excitation frequencies of acoustic waves in specific physical conditions (pressure, acceleration, temperature, humidity, etc.) [2,3,4,5,6].

Recently, the development of gas sensor acoustoelectronic devices is associated with the use of 1D (ZnO and AlN nanorods) and 2D crystals (graphene and graphene oxide) [7,8,9,10]. One-dimensional and 2D crystals on the surface of piezoelectric crystals act as adsorbents, which leads either to a change in the electrical properties of 1D and 2D crystals or to a change in their mass, which ultimately leads to a change in the velocity and excitation frequency of surface and bulk acoustic waves in crystals.

An interesting proposal is to use the surface acoustic waves in solar energy for acoustically stimulated transport of charge carriers generated in semiconductor structures under the influence of light [11,12,13,14]. In the future, the use of SAW in solar cells can increase the efficiency of cells up to 90% [14]. SAWs propagating in piezoelectric crystals (including GaN and GaAs crystals) have opposite potential values in the acoustic wave minima and maxima due to the piezoelectric effect. Electrons and holes generated in the semiconductor structure under the influence of sunlight are distributed, respectively, between the minima and maxima of the SAW. The charges are then carried by the SAW to the solar cell output at the velocity of an acoustic wave. The use of SAW in solar cells can increase the charge collection area in the semiconductor structure and increase the efficiency of solar cells. Perspectives in solar energy are also associated with the use of graphene [15,16], which can be used as transparent electrodes, and in which electron-hole pairs can also be generated under the influence of light. Moreover, the graphene film is a good medium for SAW propagation. A number of studies have demonstrated the possibility of charge transport by the surface acoustic wave in graphene [17,18,19], as well as the possibility of controlling the acoustic wave under the application of an electric potential to the graphene film [20]. In [21], the possibility of fabrication of the interdigital transducer (IDT) from graphene for SAW excitation was demonstrated. Here, the interest in the application of graphene is due to its association with its practically zero mass and the absence of influence on the SAW propagation. In [22], using the Talbot effect, the SAW propagation process was visualized and the dependence of the current in the graphene film on the SAW amplitude was studied. Also, the process of acoustically stimulated charge transport in the graphene film was visualized by scanning electron microscopy and the induced current method [23]. A number of studies have demonstrated the transport of charges generated by sunlight in the graphene film with the help of the SAW [24,25].

The purpose of these investigations is to study the process of the SAW propagation in the graphene film on the surface of a LiNbO_3_ ferroelectric crystal depending on the SAW amplitude and the bias potential on the graphene film. The process of charge transport in a graphene film under the conditions of generation of electron-hole pairs under the action of optical radiation is also studied.

## 2. Fabrication of Graphene/LiNbO_3_ SAW Delay Time Line

The YZ−cut of a LiNbO_3_ crystal was used to study the process of the SAW propagation in the graphene film. This crystal cut has large values of piezoelectric moduli and a large value of the electromechanical coupling coefficient. The process of SAW propagation in the YZ−cut of a LiNbO_3_ crystal is characterized by the autocollimation of the acoustic beam. The velocity of SAW propagation in the YZ−cut of the LiNbO_3_ crystal is V=3420 m/s. The well-known value of the SAW velocity in the YZ−cut of a LiNbO_3_ crystal is V=3488 m/s, but the actual value of the velocity depends on the particular crystal and can vary. The substrate size was 8 × 16 × 1 mm^3^. The substrate surface has been polished. The substrate roughness does not exceed 3 Å at the damaged layer thickness of 24 Å.

A SAW delay time line consisting of two identical interdigital transducers (IDTs) was fabricated on the surface of the YZ−cut of a LiNbO_3_ crystal. The IDTs (input and output) were fabricated by photolithography and consisted of 25 pairs of Al electrodes. The period of the IDT electrodes was 15 μm, which corresponds to the SAW wavelength Λ = 30 μm. The aperture of the IDT was *W* = 50 × Λ = 1500 µm.

Figure 1 shows an optical micrograph of the SAW delay time line. A graphene film was formed on the substrate surface between two IDTs by a transfer method. Figure 2 shows the Raman spectra of the graphene film transferred to the surface of the YZ−cut of a LiNbO_3_ crystal. In the left part of the spectrum, strong peaks of the LiNbO_3_ crystal are visible. The inset shows a zoomed-in part of the spectrum, in which the peaks of graphene (G, 2D, D+D”, 2D’) are visible. The spectrum demonstrates features characteristic of single-layer graphene, namely, a symmetric 2D peak with a full width at half-maximum FWHM = 28 cm^−1^ and intensity ratio I_2D_/I_G_ = 2.1 [26,27]. In the inset of Figure 2, the peaks related to the Raman spectrum from the LiNbO_3_ substrate are marked by asterisks.

To study the acoustically stimulated charge transport on the graphene surface, four gold electrodes were fabricated (Figure 1). The distance between the electrodes was 800 µm. The two central electrodes were used to measure the current I in the graphene film; the distance between them was 800 µm. The two outermost electrodes were used to apply the bias potential U and the distance between them was 2400 µm.

## 3. Investigation of SAW Propagation in Graphene Film by Scanning Electron Microscopy Method

Usually, to study the operation of acoustoelectronic devices (in our case the SAW delay time line), the amplitude-frequency characteristics are measured, which compare the electrical signals at the input and output interdigital transducers. Figure 3 shows the amplitude-frequency response S21 (acoustic wave transmission coefficient from input IDT to output IDT) of the SAW delay time line. The figure clearly shows that the resonance excitation frequency of the SAW is $f_{0}=114.5$ MHz, and the dependence of the amplitude-frequency response itself has the form of a sinx/x function. Unfortunately, the amplitude-frequency response does not provide information about the real SAW propagation on the surface of the piezoelectric substrate.

To study the process of the SAW propagation on the surface of a piezoelectric substrate with graphene, a scanning electron microscopy method was used to visualize the SAW in real time [28,29,30]. The process of the SAW propagation in the SAW delay time line was investigated using a JEOL JSM840 SEM at an electron probe current of 1 nA and an accelerating voltage of 1 kV. The use of higher accelerating voltages of electrons leads to the formation of negative charges on the surface of the piezoelectric substrate and distortion of the formed image. The SAW image is formed in the mode of recording of low-energy secondary electrons with energy of ~1 eV, which are sensitive to the piezoelectric potential that accompanies the SAW propagation on the surface of the piezoelectric substrate.

Figure 4 shows the SEM microphotographs of SAW propagation in the SAW delay time line, the surface of which in the central part is covered by a film of single-layer graphene. Figure 4a shows the SEM microphotograph of the traveling SAW excited by the input IDT at the resonant excitation frequency of $f_{0}=114.5$ MHz. In an acoustic beam at the output of the IDT, the Fresnel diffraction pattern can be observed that is associated with the SAW diffraction at the IDT aperture (analogous to the diffraction of optical radiation on a slit). The structure of the acoustic wave field of SAW in the graphene film is observed much weaker compared to the free surface of the crystal. Figure 4b shows a SAW image in the graphene film and at the output from the graphene film. The decrease of the SAW contrast in the graphene film is due to the fact that the graphene film has a small conductivity and acts as an equipotential surface, which screens the piezoelectric potential of the traveling SAW in the crystal. Since the resistance of the graphene film is several kΩ, the SAW image can be observed on the surface of the graphene film.

## 4. Investigation of Acoustically Stimulated Charge Transport in Graphene Film

To study the process of charge transport in the graphene film by the surface acoustic wave, the scheme shown in Figure 1 was applied. Two central Au electrodes were used to measure the current in the graphene film under the conditions of the SAW propagation. The two outermost Au electrodes were used to apply a bias voltage U to the graphene film. The bias voltage was varied in the range of U=−80 to 80 mV. The studies were carried out in the frequency range of f=95 to 135 MHz, which corresponds to the frequency range of the amplitude-frequency response in Figure 3.

Figure 5 shows the dependences of the current in the graphene film as a function of the amplitude of the input signal on the IDT. The amplitude of the input signal on the IDT was varied from 0 to 20 V. The current in the graphene film was measured in the frequency range of f=95÷135 MHz. The structure of the frequency dependence of the current in the graphene film is similar to the structure of the amplitude-frequency response and has the form of a sinx/x function. Small oscillations on the current-frequency dependences (Figure 5) and on the amplitude-frequency response of the SAW delay time line (Figure 3) are determined by the electrode’s structure of the interdigital transducer (number of electrodes pairs). The dependences of the current on frequency clearly show that with an increase in the SAW amplitude, the current in the graphene film increases. At a negative bias U=−80 mV (Figure 5a) in the graphene film, a current is already observed between two gold electrodes in the absence of the SAW propagation, and SAW excitation leads to an increase in the current between the electrodes. At zero bias U=0 mV (Figure 5b) the current in the graphene film is zero and begins to increase with increasing amplitude of the input signal on the IDT. With a positive bias U=80 mV (Figure 5c) in the absence of SAW, the current between the electrodes has the opposite direction with respect to the current in Figure 5a. In this case, SAW propagation leads to a decrease in current; at certain amplitudes of the input signal at the IDT, the current becomes equal to zero, since the charge transport due to the positive bias U=80 mV is compensated by the charge transport by the surface acoustic wave in the opposite direction. A further increase in the amplitude of the SAW, by increasing the amplitude of the input signal on the IDT, leads to an increase in current in the opposite direction with respect to the initial current direction. That is, in this case, with a positive bias U=80 mV at the SAW resonance excitation frequency of $f_{0}=114.5$ MHz the SAW propagation in the YZ−cut of a LiNbO_3_ crystal allows to change the direction of charge transport in the graphene film. In Figure 5a,b at a negative and zero bias potential, the current has a constant direction in the frequency range of f=95÷135 MHz, while at a positive bias potential U=80 mV (Figure 5c), the current direction in the graphene film depends on the SAW excitation frequency and the SAW amplitude. Therefore, in this case, by varying the SAW excitation frequency within the amplitude-frequency response, it is possible to change the direction of the current in the graphene film.

Figure 6 shows the dependences of the current in the graphene film on the amplitude of the input signal at the IDT (dB), measured at the bias potentials U=−80, 0, 80 mV and the SAW resonance excitation frequency of $f_{0}=114.5$. The dependence of the current in the graphene film on the amplitude of the input signal at the IDT in dB has a linear character. At a bias potential of U=80 mV and in the absence of SAW propagation, the current in the graphene film is I=100 nA. Excitation of the SAW leads to the beginning of the charge transport process in the graphene film in the opposite direction. When the input signal amplitude at the IDT is 7 dB, the charge transport process is compensated and the current in the graphene film becomes equal to zero. A further increase in the SAW amplitude by increasing the amplitude of the high-frequency input signal at the IDT leads to an increase in current in the opposite direction. At the amplitude of the input signal at the IDT 100 dB, the current in the graphene film is I=−1470 nA. Thus, it has been demonstrated that the SAW allows the charge transport in the graphene film, and this process depends on the amplitude of the SAW on the surface of the substrate. An increase in the SAW amplitude leads to an increase in the current in the graphene film.

Hence, it is possible to use the surface acoustic waves to control the direction and magnitude of the current in a graphene film by changing the SAW amplitude or changing the SAW excitation frequency.

## 5. Transport of Photo-Stimulated Charges in a Graphene Film by Surface Acoustic Waves

Acoustically stimulated charge transport is of interest for the development of solar cells, where using of the surface acoustic waves allows collecting charges from large surfaces [11,12,13,14]. Figure 7 shows a diagram of a simulated experiment to study the transport of photo-stimulated charges by the surface acoustic wave. A laser with a wavelength of 532 nm and a power of 5 mW was used as a light source. The size of the laser beam spot on the surface of the YZ−cut of a LiNbO_3_ crystal covered by the graphene film was 1 mm in diameter, which is less than the aperture of the IDT W=1500 μm.

Charge carriers are generated under the influence of laser radiation in the near-surface area of the LiNbO_3_ crystal. The use of graphene as a conductive medium makes it possible to move the photo-stimulated charges by surface acoustic wave.

The bias potential of U=−80 mV was used for the investigation of the transport of photo-stimulated charges. The SAW excitation frequency was $f_{0}=114.5$ MHz at an input high-frequency signal power of 100 dB. Illumination of the crystal surface by the laser radiation allows increasing the value of current I in the graphene film between the two central gold electrodes (Figure 1 and Figure 7).

Figure 8 shows the dependences of the current in the graphene film under the on/off switching of illumination of the YZ−cut of a LiNbO_3_ crystal surface in the absence and under the excitation of the surface acoustic waves. In the absence of the surface acoustic waves, the current in the graphene film at the bias potential of U=−80 mV was I=−100 nA. Illumination of the YZ−cut of a LiNbO_3_ crystal with a graphene film leads to an increase in the current in the graphene film to I=−170 nA. The propagation of surface acoustic waves in the LiNbO_3_ crystal leads to a significant increase in the current in the graphene film. Thus, in the absence of illumination, the current value in the graphene film is I=−1800 nA. Illumination of the crystal surface leads to a significant increase in the current in the graphene film up to I=−2850 nA. Figure 8 shows the current modulation under conditions of on/off switching of illumination of the crystal surface by the laser radiation source. Thus, the acquisition and transport of charge carriers in the graphene film by the surface acoustic wave was demonstrated.

In the future, this approach allows increasing efficiency of the solar cells by increasing the collection of charge carriers with the use of surface acoustic waves.

## 6. Conclusions

The process of acoustically stimulated charge transport in the graphene film on the surface of the YZ−cut of a LiNbO_3_ crystal was investigated. It is shown that the frequency dependence of the current in the graphene film repeats the amplitude-frequency response of the SAW delay time line. It is demonstrated that the increase of the SAW amplitude (the increase of the input signal amplitude on the IDT) leads to an increase in the current in the graphene film. The dependence of the current in the graphene film depends linearly on the amplitude of the input signal on the IDT in dB.

At a positive bias potential U=80 mV on the graphene film, the propagation of the SAW can change the direction of the current in the graphene film due to a change in the SAW amplitude. At the certain amplitude of the SAW (the amplitude of the input signal on the IDT is 3.5 dB), the current in the graphene film becomes equal to zero. It should also be noted that in the frequency range of the amplitude-frequency response, the current in the graphene film can vary from positive to negative values depending on the frequency. Thus, the possibility of controlling the value and direction of the current in the graphene film by changing the frequency of the SAW excitation or changing the amplitude of the SAW has been demonstrated.

The possibility of the transport of the photo-stimulated charges by the surface acoustic wave, i.e., the charges generated by the optical radiation in the ferroelectric substrate of the YZ−cut of a LiNbO_3_ crystal, has been demonstrated. The SAW propagation allows the collecting of the photo-stimulated charges and transferring them in the graphene film on the substrate surface. This method of charge collection and transfer will significantly improve the efficiency of solar cells in the future.

## Figures and Tables

**Figure 1 nanomaterials-12-04370-f001:**
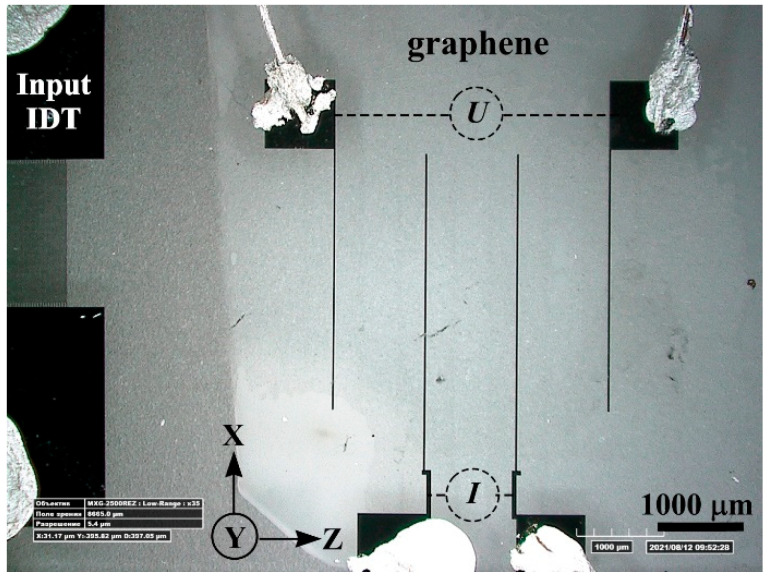
Optical image of the SAW delay time line with graphene.

**Figure 2 nanomaterials-12-04370-f002:**
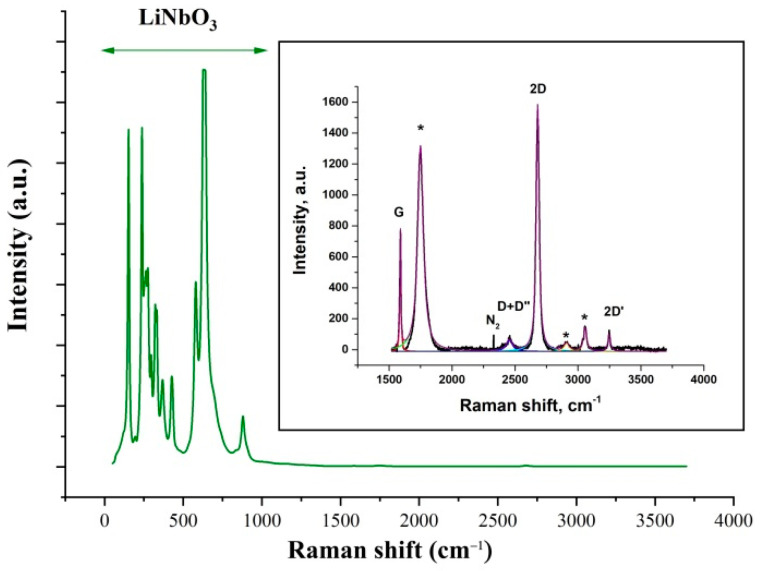
The representative Raman spectra of the graphene film transferred to the LiNbO_3_ substrate. Insert: zoomed-in spectrum showing graphene Raman peaks.

**Figure 3 nanomaterials-12-04370-f003:**
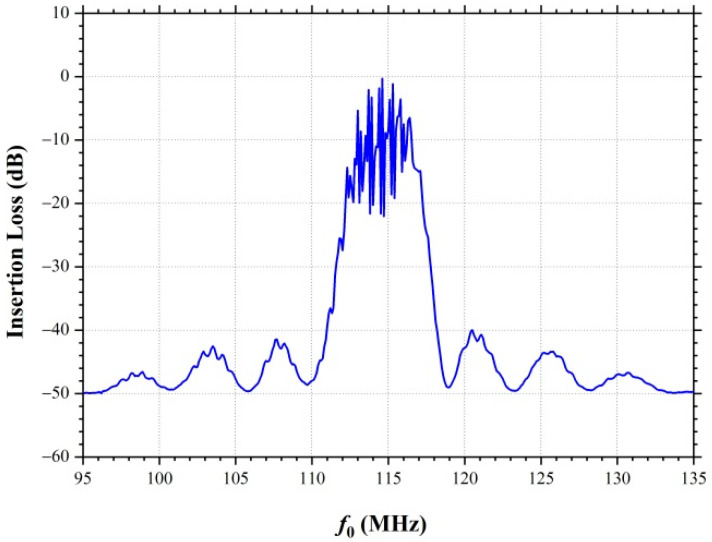
Electrical characterization S21 of the SAW delay time line.

**Figure 4 nanomaterials-12-04370-f004:**
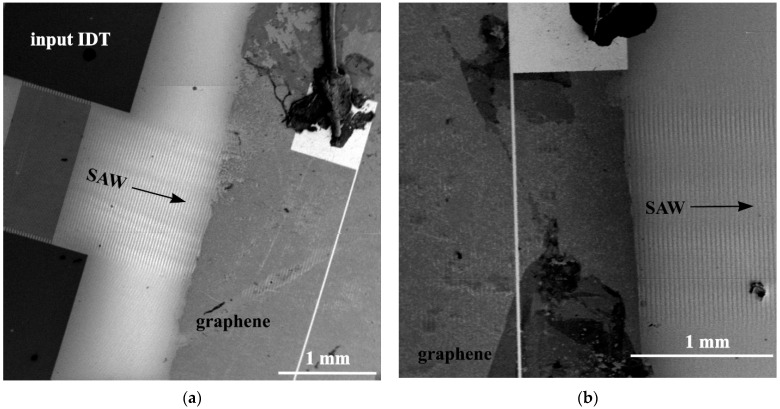
SAW propagation in the YZ−cut of a LiNbO_3_ crystal covered by graphene film: (**a**) SAW excitation by interdigital transducer, (**b**) SAW propagation in the graphene film and on the free crystal surface. Λ=30 µm, $f_{0}=114.5$ MHz, V=3435 m/s.

**Figure 5 nanomaterials-12-04370-f005:**
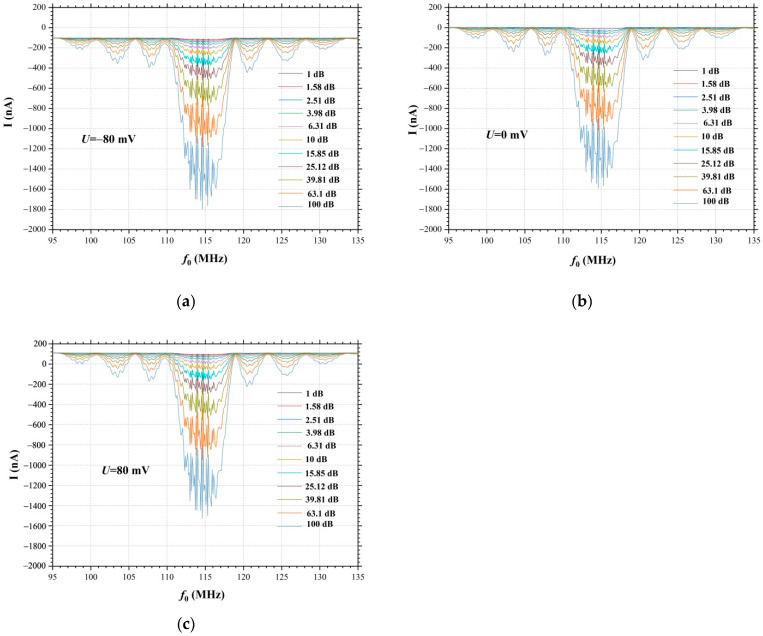
Dependences of the current in the graphene film on the bias potential *U* and the amplitude of the input signal supplied to the IDT (1÷100 dB): (**a**) U=−80 mV, (**b**) U=0 mV, (**c**) U=−80 mV.

**Figure 6 nanomaterials-12-04370-f006:**
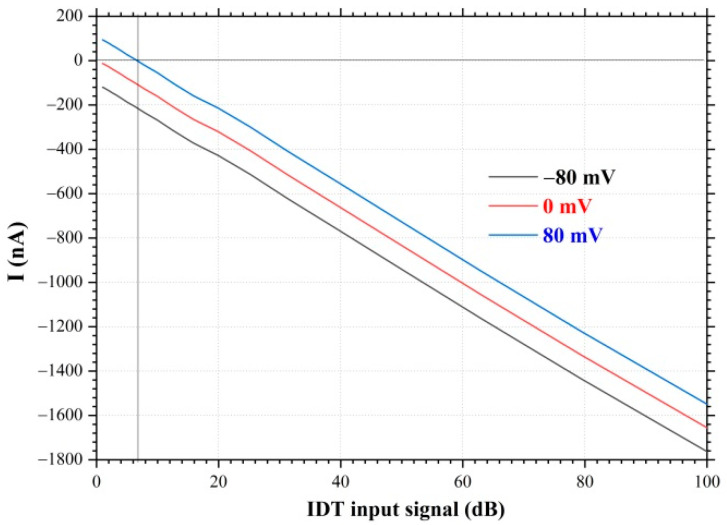
Dependences of current in graphene film versus amplitude of the input signal supplied to IDT (in dB). YZ−cut of a LiNbO_3_ crystal, Λ=30 µm, $f_{0}=114.5$ MHz, V=3420 m/s.

**Figure 7 nanomaterials-12-04370-f007:**
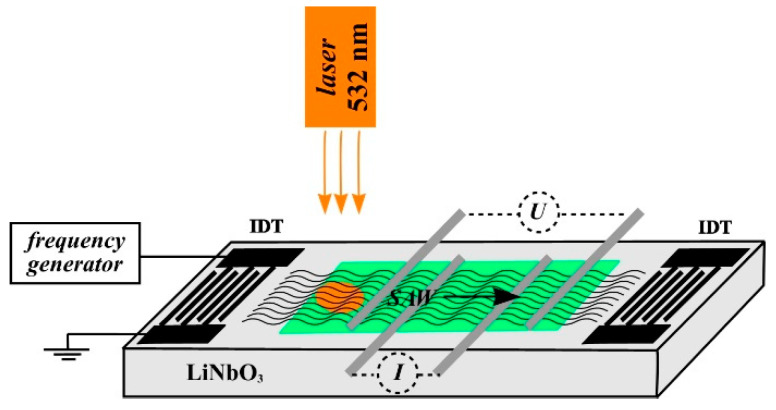
Scheme of a photo-stimulated charge transport in a graphene film by a surface acoustic wave.

**Figure 8 nanomaterials-12-04370-f008:**
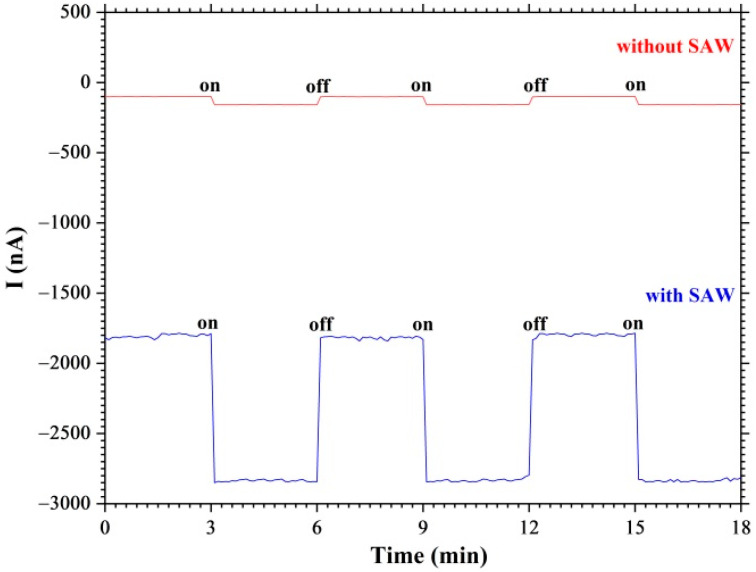
Dependences of acoustically stimulated photocurrent in graphene film on the on/off switching of the surface acoustic wave.

## Data Availability

All relevant data presented in the article are stored according to institutional requirements and as such are not available online. However, all data used in this manuscript can be made available upon request to the authors.

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
