# Peer review of "Acoustically Stimulated Charge Transport in Graphene Film"

_nanomaterials, 2022, doi:10.3390/nano12244370_

Round 1

Reviewer 1 Report

This manuscript studied the acoustically stimulated charge transport in the graphene film. The current response depending on the SAW frequency and amplitude was investigated. The author also demonstrated photostimulated charges transport was also demonstrated in the experiment. The manuscript is well drafted with comprehensive experiment and analysis. I would recommend publication after the following revision:

1.     About the transducer configuration, the structure of input IDT was well presented. But the structure of the output IDT was not clearly illustrated. Which two electrodes has a distance of 800um? What’s the distance between the two bias electrodes and the two current electrodes?

2.     How is the S21 of the SAW delay line measured? Which electrodes were used as output IDT in this measurement?

3.     Minor:

a.     Some unit missing: page 6 line 187, f0=114.5MHz; page 9 line 252, U=80mV

b.     Page 5 line 146: U= -80 to 80 mV? f = 95 to 135 MHz?

c.     Page 5 line 150: “The amplitude of the input signal on the IDT was varied from 1 dB to 100 dB”, what does this amplitude reference to? dB is not a relative unit.

Author Response

Dear reviewer,

Thanks so much for your helpful comments.

  1. About the transducer configuration, the structure of input IDT was well presented. But the structure of the output IDT was not clearly illustrated. Which two electrodes has a distance of 800um? What’s the distance between the two bias electrodes and the two current electrodes?

Lines 79-80: A SAW delay time line consisting of two identical interdigital transducers (IDTs) was fabricated on the surface of the cut of a LiNbO3 crystal. Two IDTs (input and output)

Lines 100-101: The two central electrodes were used to measure the current  in the graphene film, the distance between them was 800 µm. The two outermost electrodes were used to apply the bias potential  and the distance between them was 2400 µm, respectively.

  1. How is the S21 of the SAW delay line measured? Which electrodes were used as output IDT in this measurement?

Line 107: (acoustic wave transmission coefficient from input IDT to output IDT)

  1. Minor:
  2. Some unit missing: page 6 line 187, f0=114.5MHz; page 9 line 252, U=80mV

Line 187:  MHz

Line252: At a positive bias potential  mV on the graphene film, the propagation of the

  1. Page 5 line 146: U= -80 to 80 mV? f = 95 to 135 MHz?

Lines 146-147: bias voltage was varied in the range of  mV. The studies were carried out in the frequency range of  MHz, which corresponds to the frequency range

  1. Page 5 line 150: “The amplitude of the input signal on the IDT was varied from 1 dB to 100 dB”, what do

Line 150: IDT was varied from 0 to 20 V

Reviewer 2 Report

I read the manuscript "Acoustically stimulated charge transport in graphene film" with interest. The results seem solid, but the way of presenting data is sometimes not very nice. I would like to suggest the following points before recommending publication.

1. YZ-cut should be explained in the figure.

2. The length scale should be explicitly shown in Fig. 1.

3. As for Fig. 2, "features characteristic of single-layer graphene" needs a reference.

4. The meaning of parameter S21 should be explained.

5. In Fig. 7, the wavelength of 532 nm should be explicitly written. Why do the authors use 532 nm for this purpose?

6. In the abstract, "IDT" is not defined. 

7. In the conclusion, the parameter "U" should have a unit of V or mV.

Author Response

Dear reviewer,

Thanks so much for your helpful comments.

  1. YZ-cut should be explained in the figure.

We made corrections and indicated the direction of axes in the YZ-cut of the LiNbO3 crystal in Figure 1.

  1. The length scale should be explicitly shown in Fig. 1.

The length scale is shown in Figure 1.

  1. As for Fig. 2, "features characteristic of single-layer graphene" needs a reference.

The following references have been inserted:

  1. Ferrari, A. C.; Basko, D. M. Raman spectroscopy as a versatile tool for studying the properties of graphene. Nat. Nanotechnol. 2013, 8, 235–246.
  2. Ferrari, A. C.; Meyer, J.C.; Scardaci, V.; Casiraghi, C.; Lazzeri, M.; Mauri, F.; Piscanec, S.; Jiang, D.; Novoselov, K.S.; Roth, S.; Geim, A.K. Raman spectrum of graphene and graphene layers. Phys. Rev. Lett. 2006, 97, 187401.
  3. The meaning of parameter S21 should be explained.

Lines 107: (acoustic wave transmission coefficient from input IDT to output IDT)

  1. In Fig. 7, the wavelength of 532 nm should be explicitly written. Why do the authors use 532 nm for this purpose?

The 532 nm wavelength is tied to the laser in Figure 7.

We used the laser that we had at our disposal. In the long term, it is interesting to conduct studies for different wavelengths of optical radiation as well as for different cuts of the LiNbO3 crystal. There are a number of cuts that have a higher electromechanical coupling coefficient and, correspondingly, a higher charge transport efficiency. But this is research for the future.

  1. In the abstract, "IDT" is not defined. 

Line 15: interdigital transducer (IDT, in dB).

  1. In the conclusion, the parameter "U" should have a unit of V or mV.

Line 252: At a positive bias potential  mV on the graphene film, the propagation of the
